# Comparison of Tomographic Findings in Patients with Influenza A (H1N1) and COVID-19

**DOI:** 10.3390/diagnostics15111430

**Published:** 2025-06-04

**Authors:** Lourdes Noemí Vélez-Ramírez, Omar Jiménez-Zarazúa, Luis Ernesto González-Najera, Gustavo Adolfo Flores-Saldaña, Adolfo Valdez-Escobedo, Jaime Daniel Mondragón

**Affiliations:** 1Department of Radiology, Hospital General León, León 37672, Guanajuato, Mexico; nestlynoemi@gmail.com (L.N.V.-R.); legn2590@hotmail.com (L.E.G.-N.); drgustavofloresrx@hotmail.com (G.A.F.-S.); 2Department of Medicine and Nutrition, Universidad de Guanajuato, León 37320, Guanajuato, Mexico; 3Department of Internal Medicine, Hospital General de Zona IMSS No. 21, León 37380, Guanajuato, Mexico; 4Department of Epidemiology, Hospital General León, León 37672, Guanajuato, Mexico; avaldeze@guanajuato.gob.mx; 5Department of Neurology, University Medical Center Groningen, University of Groningen, 9713 LG Groningen, The Netherlands; 6Laboratorio de Psicofisiología, Departamento de Neurobiología Conductual y Cognitiva, Instituto de Neurobiología, Universidad Nacional Autónoma de México, Querétaro 76230, Querétaro, Mexico; 7Life-Span Human Senses Lab, Department of Psychology, San Diego State University, San Diego, CA 92182, USA

**Keywords:** influenza A virus subtype H1N1, artificial respiration, mortality, SARS-CoV-2 (COVID-19), organ dysfunction scores

## Abstract

**Objectives**: The objective of this study was to identify CT-based predictors of mechanical ventilation and mortality in patients with severe and critical viral pneumonia and to examine the association between imaging severity and outcomes in ventilated patients. **Methods**: We analyzed pulmonary CT scans from 148 patients with severe or critical pneumonia caused by COVID-19 (*n* = 98) or influenza A H1N1 (*n* = 50). Patients were assessed based on tomographic patterns, demographics, clinical severity scores (Charlson Comorbidity Index, SOFA, and APACHE IV), and biomarkers. Survival analyses were performed using Kaplan–Meier curves and multivariable Cox regression. **Results**: Bilateral, peripheral, and basal lung involvement was common across both groups. Ground-glass opacities (89.62%, *p* ≤ 0.001) and consolidation (61.54%, *p* = 0.001) were more prevalent in COVID-19, whereas pleural effusion was significantly more frequent in H1N1 (76.92%, *p* ≤ 0.001). COVID-19 cases more often presented with bilateral (96.94%) and peripheral lesions (77.87%). H1N1 patients were more likely to develop severe ARDS and require mechanical ventilation. In COVID-19, higher APACHE IV scores and pulmonary damage severity index were independently associated with increased mortality. **Conclusions**: Radiologic and clinical severity profiles differ between COVID-19 and H1N1 pneumonia. CT-based assessments combined with prognostic scores may aid early risk stratification and guide treatment decisions in patients with severe viral pneumonia.

## 1. Background

Severe acute respiratory syndrome coronavirus 2 (SARS-CoV-2), the causative agent of COVID-19, was declared a global pandemic by the World Health Organization (WHO) on 11 March 2020 [1]. More than a decade earlier, influenza A H1N1 was similarly recognized as a pandemic in June 2009 [1]. Viral pneumonia remains a major global health challenge due to its high burden of morbidity and mortality, with an estimated 200 million community-acquired cases annually—affecting children and adults equally [2]. Although both SARS-CoV-2 and influenza A H1N1 can result in severe respiratory illness, their mortality profiles differ. The global mortality rate for H1N1 in 2009 was estimated at approximately 11%, rising to 18–20% among individuals over the age of 50 [3], while critically ill COVID-19 patients requiring intensive care have demonstrated a mortality rate around 4.3% [4]. Diagnosis of both infections relies on clinical presentation of respiratory tract symptoms in conjunction with laboratory confirmation via reverse transcriptase–polymerase chain reaction (RT-PCR) testing [5,6].

### 1.1. Imaging in COVID-19 and Influenza

A variety of radiologic manifestations have been associated with infections caused by SARS-CoV-2 and influenza viruses. While imaging is not definitive for diagnosis, pattern recognition on chest scans can assist clinicians in distinguishing viral pneumonias from bacterial ones, thereby supporting more targeted antimicrobial use [7]. Research linking specific imaging findings to viral lower respiratory tract infections remains limited [8]. Nevertheless, computed tomography (CT) has proven particularly useful in evaluating suspected cases of pneumonia due to its enhanced sensitivity compared to other modalities [9]. In COVID-19, CT frequently reveals multifocal ground-glass opacities with irregular zones of consolidation, most often located in the lung periphery or within the posterior and lower lobes [10,11,12]. Diagnostic challenges persist, however, as a subset of patients—particularly those with milder or early-stage disease—may show no abnormalities on CT at the time of hospital presentation [13]. Comparative analyses have highlighted several distinctions between COVID-19 and other viral pneumonias; for instance, COVID-19 cases tend to show greater peripheral distribution (80% vs. 57%), more ground-glass opacities (91% vs. 68%), and increased fine reticular and vascular patterns. In contrast, infections caused by other respiratory viruses more often demonstrate a mixed pattern, higher rates of pleural effusion (39% vs. 4.1%), and more frequent lymphadenopathy (10.2% vs. 2.7%) [11]. In a separate study comparing COVID-19 and influenza-related pneumonia, findings such as round opacities and interlobular septal thickening were more prevalent in COVID-19, whereas influenza cases were more likely to exhibit nodular lesions, dense micronodules, and effusions [12].

### 1.2. Objective

This study presents a prospective comparison of pulmonary CT findings in patients diagnosed with severe or critical pneumonia caused by either SARS-CoV-2 or influenza A H1N1. Our primary aim was to differentiate the imaging characteristics between these two viral infections by analyzing demographic profiles, tomographic features, and prognostic markers. We systematically categorized patients according to radiological features, disease severity scores, and biomarker profiles to explore associations with clinical outcomes.

A secondary aim was to assess the likelihood of requiring invasive mechanical ventilation and to evaluate whether higher disease severity correlates with increased mortality. Using imaging-based indices and clinical severity metrics, we examined how lung involvement and ARDS classification may predict outcomes in patients who underwent mechanical ventilation.

## 2. Study Design and Methods

### 2.1. Population

This was a prospective observational cohort study with consecutive enrollment of eligible patients admitted for severe or critical respiratory symptoms, carried out at Hospital General Regional in León, Mexico, encompassing patients diagnosed with pneumonia caused by either SARS-CoV-2 or influenza A H1N1 between June 2020 and May 2021, a period corresponding to the first and second waves of the COVID-19 pandemic in Mexico, during which ancestral variants of SARS-CoV-2 predominated prior to the emergence of Omicron-related strains. All CT scans were performed within 24 h of admission, and clinical and laboratory data were collected prospectively using standardized data abstraction protocols developed prior to patient enrollment. Viral infection was confirmed via reverse transcriptase–polymerase chain reaction (RT-PCR). Eligible participants were adults (≥18 years) admitted through the emergency department following clinical assessment indicating respiratory compromise, oxygen supplementation needs, hemodynamic instability, or other acute indications for hospitalization.

All participants underwent chest CT within 24 h of PCR sample collection. Clinical severity (severe vs. critical pneumonia) was determined based on symptoms, laboratory markers, and imaging findings. Upon admission, standardized treatment protocols appropriate to the identified viral etiology were initiated.

Inclusion criteria required a completed 30-day medical follow-up and comprehensive clinical documentation. Exclusion criteria comprised individuals with lung malignancies, metastatic disease, autoimmune disorders (e.g., lupus, rheumatoid arthritis), or preexisting interstitial lung conditions. Patients who declined CT imaging, did not consent to data use, or lacked follow-up data were excluded from final analyses. Institutional Review Board approval was obtained (IRB #00743-20), and the study was registered on ClinicalTrials.gov (NCT04497311), in accordance with the Declaration of Helsinki.

### 2.2. Image Acquisition and Diagnosis

Chest CT scans were acquired using a Siemens 16-slice scanner that had been upgraded to 64-slice capacity. Pixel resolution was set at 0.72 mm for the uCT platform and 0.85 mm for the Siemens system (Erlangen, Germany), with a uniform slice thickness of 5 mm across both devices. All imaging was reviewed independently by two radiologists.

At admission, nasopharyngeal swabs were obtained for both SARS-CoV-2 and influenza A H1N1 detection. COVID-19 testing utilized the SuperScript III One-Step RT-qPCR system (Invitrogen, Thermo Fisher Scientific, Waltham, MA, USA), while H1N1 testing followed World Health Organization (WHO) and Centers for Disease Control and Prevention (CDC) protocols. Influenza RNA was extracted and quantified using the PureLink Viral RNA/DNA Mini Kit (Thermo Fisher Scientific, Waltham, MA, USA).

### 2.3. Data Extraction

Patient data were systematically retrieved from clinical records, including demographic characteristics such as age, sex, weight, height, and body mass index (BMI). Clinical indicators recorded included total hospitalization duration, in-hospital progression, baseline vital signs, and documented comorbidities such as cardiac failure, renal impairment, and respiratory insufficiency.

Key laboratory values were also extracted, including serum lactate levels, arterial blood gases (PaO_2_/FiO_2_), inflammatory markers (e.g., D-dimer, CRP, and LDH), and hematologic and metabolic parameters (platelet count, leukocyte count, hemoglobin, liver enzymes, creatinine, and albumin). Treatment interventions during hospitalization were cataloged, including mechanical ventilation, vasopressor or inotropic support, corticosteroid use, antibiotic therapy, and renal replacement techniques when applicable.

Three prognostic scoring systems were used to assess clinical severity and risk, i.e., the Charlson Comorbidity Index (CCI), the Sequential Organ Failure Assessment (SOFA), and the Acute Physiology and Chronic Health Evaluation IV (APACHE IV). CCI reflects preexisting comorbidity burden, while SOFA and APACHE IV quantify acute physiological dysfunction and severity of illness at admission. These scores were used as independent variables in logistic regression to predict mechanical ventilation and in Cox regression models to evaluate mortality risk.

### 2.4. Patient Classification and Radiological Assessment

Pneumonia severity was determined using established clinical criteria. Patients were categorized as having severe pneumonia if they exhibited respiratory distress—defined as a respiratory rate greater than 30 breaths per minute—oxygen saturation below 93% at rest, or a PaO_2_/FiO_2_ ratio under 300 mmHg [14]. Those requiring mechanical ventilation, presenting with circulatory shock, or experiencing multiple organ dysfunction requiring admission to the intensive care unit (ICU) were classified as having critical pneumonia [14].

Radiologic evaluations were independently performed by two board-certified radiologists blinded to clinical and laboratory information. Agreement between observers was assessed using Cohen’s kappa (κ = 0.84). Any discrepancies in interpretation were resolved by consensus following a review by a third radiologist. All radiological assessments were conducted blinded to clinical and laboratory data.

All chest CT scans were assessed for characteristic imaging features (Figure 1), including consolidation, ground-glass opacities, nodular lesions, lymphadenopathy, vascular thickening, and evidence of embolism (Figure 2). To quantify lung involvement, we used a semi-quantitative pulmonary damage severity index based on estimated parenchymal involvement in each of the five lung lobes. Each lobe was scored from 0 (no involvement) to 5 (more than 75% involvement), yielding a total score ranging from 0 to 25 per patient. This approach, previously validated in similar contexts [15,16], was selected over standardized diagnostic scoring systems such as CO-RADS or RALE due to its flexibility and broader applicability in comparative viral pneumonia analysis. Lymphadenopathy was defined as lymph nodes with a short-axis diameter greater than 1 cm. Vascular thickening was identified by abnormal vessel enlargement—either greater than nearby vessels in unaffected areas, larger than corresponding vessels in the contralateral lung, or demonstrating a lack of tapering toward the periphery.

The anatomical distribution of pulmonary lesions was also recorded, differentiating between unilateral or bilateral involvement, peripheral versus central localization, and predominance in the basal lobes. A pulmonary damage severity index was calculated based on estimated parenchymal involvement in each of the five lung lobes. Each lobe was assigned a score ranging from 0 (no involvement) to 5 (greater than 75% involvement), yielding a cumulative score from 0 to 25 per patient [15,16].

### 2.5. Statistical Analysis

All statistical analyses were performed using SPSS version 26 (IBM Corp., Armonk, NY, USA). The dataset was screened for missing values and outliers and tested for normality assumptions through the Shapiro–Wilk test and visual inspections using histograms and Q-Q plots. Descriptive statistics were used to summarize clinical and demographic variables, reported as frequencies, percentages, means, or medians, as appropriate.

For categorical variables, comparisons between groups (COVID-19 vs. H1N1) were evaluated using chi-square (χ^2^) tests or Fisher’s exact tests when expected counts were low. Continuous variables such as age, BMI, hospital stay, and prognostic scores (e.g., APACHE IV, SOFA, CCI, and pulmonary damage severity index) were analyzed using independent-sample ANOVA. A Bonferroni correction was applied to adjust for multiple comparisons, setting the significance threshold at *p* < 0.0056. Effect sizes for ANOVAs were calculated using eta squared (η^2^).

A binary logistic regression model with backward stepwise elimination was used to identify predictors of invasive mechanical ventilation. The model included APACHE IV, SOFA, CCI, and pulmonary damage severity index. Interaction terms such as APACHE IV × SOFA and APACHE IV × CCI were tested to evaluate whether combined physiological stress and organ dysfunction or comorbidity burden had synergistic effects on outcomes. Backward stepwise elimination was employed to refine the model, using −2 log likelihood and pseudo-R^2^ statistics (Hosmer–Lemeshow, Cox–Snell, and Nagelkerke) to assess fit. Odds ratios (OR) and 95% confidence intervals (CI) were reported for each variable.

Survival analyses were conducted using the Kaplan–Meier method. Differences in survival curves between groups were evaluated using the Gehan–Breslow–Wilcoxon test, which gives greater weight to earlier events. Additional between-group and within-group comparisons of survival distributions were performed to assess associations with mechanical ventilation and viral etiology. A multivariable Cox regression analysis with backward stepwise selection was applied to identify factors independently associated with 30-day mortality. Statistical significance was set at *p* ≤ 0.05.

## 3. Results

This study included 148 patients, of whom 91 were male. The mean age of the cohort was 56.68 years (SD ± 14.75; range: 23–98), and the average body mass index (BMI) was 28.23 (SD ± 2.87; range: 21.20–36.70). Among these patients, 98 were diagnosed with COVID-19 and 50 with influenza A H1N1, both confirmed via RT-PCR.

Radiologic evaluation identified a range of pulmonary abnormalities. Consolidation was observed in 130 cases, with a significantly higher frequency in COVID-19 patients (*n* = 80, 61.54%; *p* = 0.001). Ground-glass opacities were seen in 106 patients overall and were especially common in the COVID-19 group (*n* = 95, 89.62%; *p* ≤ 0.001). Vascular thickening was documented in 95 individuals, including 66 with COVID-19 (69.47%), though the difference between groups was not statistically significant (*p* = 0.264). Additional findings included lymphadenopathy (*n* = 71; 42 COVID-19, 59.15%; *p* = 0.082), crazy paving pattern (*n* = 42; 32 COVID-19, 76.19%; *p* = 0.108), pleural effusion (*n* = 26; 20 H1N1, 76.92%; *p* ≤ 0.001), atelectasis (*n* = 17; 15 COVID-19, 88.24%; *p* = 0.042), and the tree-in-bud pattern (*n* = 8; 6 COVID-19, 75.00%; *p* = 0.590). Pulmonary embolism was identified in a total of nine patients, including seven with COVID-19 and two with H1N1. The difference between groups was not statistically significant (*p* = 0.183). Ground-glass opacities and consolidation were significantly more common in the COVID-19 group, whereas pleural effusion predominated in H1N1 cases. Most lesions were bilateral (*n* = 133; 95 COVID-19, 96.94%; *p* ≤ 0.001), with peripheral distribution (*n* = 122; 95 COVID-19, 77.87%; *p* ≤ 0.001) and basal lobe involvement (*n* = 141; 95 COVID-19, 67.38%; *p* = 0.182). Radiological and clinical characteristics are summarized in Table 1.

Consolidation (61.54%, *p* = 0.001) and ground-glass opacities (89.62%, *p* ≤ 0.001) were more common in COVID-19, while pleural effusion (76.92%, *p* ≤ 0.001) was more common in H1N1. COVID-19 patients were also more likely to have bilateral (96.94%, *p* ≤ 0.001) and peripheral lesions (77.87%, *p* ≤ 0.001).

### 3.1. Disease Severity Prognostic Scales and Invasive Ventilation

To assess group differences in disease severity, we compared scores from four prognostic scales—APACHE IV, SOFA, Charlson Comorbidity Index (CCI), and the pulmonary damage severity index—between patients with COVID-19 and those with influenza A H1N1. Significant differences were observed in APACHE IV scores (F = 16.43, η^2^ = 0.101, *p* ≤ 0.001, 95% CI [16.03, 16.95]) and SOFA scores (F = 9.76, η^2^ = 0.063, *p* = 0.002, 95% CI [8.88, 10.17]). No significant difference was found for the CCI (F = 0.001, η^2^ = 0.000, *p* = 0.972, 95% CI [0, 2.77]), while the pulmonary damage severity index was significantly different between groups (F = 1.37, η^2^ = 0.009, *p* ≤ 0.001, 95% CI [1.07, 1.68]).

A total of 47 patients required invasive mechanical ventilation. The proportion was higher among H1N1 patients (*n* = 26), who also had a greater mean rank difference in severity scores (22.62, *p* ≤ 0.001, 95% CI [22.17, 22.98]). Following initial group comparisons, a logistic regression model was used to evaluate the predictive value of each severity score on the likelihood of requiring mechanical ventilation.

The regression analysis identified both SOFA and APACHE IV scores as significant predictors. A higher SOFA score was associated with a 6.5-fold increased likelihood of requiring mechanical ventilation (χ^2^ = 9.92, *p* = 0.002, HR = 6.464, 95% CI [2.024, 20.650]), while each one-point increase in APACHE IV was linked to a 1.3-fold increase in risk (χ^2^ = 26.996, *p* ≤ 0.001, HR = 1.292, 95% CI [1.173, 1.423]). Lower CCI scores were associated with a reduced likelihood of mechanical ventilation (χ^2^ = 5.996, *p* = 0.014, HR = 0.673, 95% CI [0.510, 0.889]). In contrast, the pulmonary damage severity index did not emerge as a statistically significant factor. Detailed coefficients and model performance metrics are provided in Table 2.

Higher APACHE IV and SOFA scores were significantly associated with increased likelihood of invasive mechanical ventilation. APACHE IV, representing acute physiological burden, and SOFA, indicating multi-organ dysfunction, each independently contributed to risk prediction. In contrast, CCI showed an inverse association with ventilation needs, suggesting that lower comorbidity burden may reflect a more acute disease severity in hospitalized cases. Patients with H1N1 were more likely (HR = 1.16) to require mechanical ventilation compared to those with COVID-19. Higher SOFA and APACHE IV scores were associated with a 6.5-fold and 1.3-fold increased likelihood, respectively, of requiring invasive mechanical ventilation.

### 3.2. Survival as a Function of the Need for Mechanical Ventilation

Following the analysis of risk factors associated with the need for invasive mechanical ventilation, we examined overall survival differences between patients who required intubation and those who did not. Survival distributions differed significantly between these groups (χ^2^ = 70.53, *p* ≤ 0.001, 95% CI [69.33, 71.17]), with lower mortality observed among patients who were not mechanically ventilated. This finding is illustrated in Figure 3.

In total, 54 patients died during the study period—29 with COVID-19 and 25 with H1N1. The median time to death for all deceased patients was 9 days (interquartile range: 4–12; range: 2–30). Among COVID-19 patients, the median time to death was also 9 days (IQR: 6–13.5), while H1N1 patients had a median of 8 days (IQR: 3–12).

The distribution of radiologic findings among deceased patients varied by etiology. In the COVID-19 group, all 29 patients had ground-glass opacities, 25 had consolidation, 21 showed vascular thickening, and 16 had lymphadenopathy. Additional findings included the crazy paving pattern in eleven patients, pleural effusion in three patients, atelectasis in five patients, the tree-in-bud pattern in one patient, and thrombosis in three patients. Bilateral and peripheral involvement were nearly universal (*n* = 28 each), and 27 showed involvement of the basal lobes.

Among deceased H1N1 patients, all 25 exhibited consolidations, but only 10 had ground-glass opacities. Vascular thickening and lymphadenopathy were observed in 16 and 15 patients, respectively. Three showed crazy paving, while pleural effusion was more common (*n* = 10). Two had atelectasis, and none showed the tree-in-bud pattern or thrombosis. Bilateral involvement was present in all cases, with peripheral and basal distribution noted in 21 and 22 patients, respectively.

To further identify predictors of 30-day mortality in ventilated patients, a Cox regression analysis was conducted using the four prognostic indices. In the full sample, only the pulmonary damage severity index was significantly associated with mortality (χ^2^ = 17.901, *p* ≤ 0.001, HR = 1.157, 95% CI [1.081, 1.238]). APACHE IV, SOFA, and CCI did not show statistically significant associations.

Survival distributions were also compared between the COVID-19 and H1N1 groups (χ^2^ = 1.592, *p* = 0.207, 95% CI [0.13, 3.47]), revealing no significant difference in overall survival by viral etiology (Figure 4A). However, when stratified by mechanical ventilation status, both diseases showed markedly higher mortality in ventilated patients.

In COVID-19 cases requiring mechanical ventilation, mortality was significantly elevated (χ^2^ = 22.60, *p* ≤ 0.001, 95% CI [21.72, 23.03]; Figure 4B). In this subgroup, multivariable Cox regression identified APACHE IV (χ^2^ = 7.124, *p* = 0.008, HR = 1.029) and the pulmonary damage severity index (χ^2^ = 16.843, *p* ≤ 0.001, HR = 1.258) as significant predictors of death. Neither SOFA (χ^2^ = 0.850, *p* = 0.356) nor CCI (χ^2^ = 1.319, *p* = 0.251) reached statistical significance.

A similar pattern was observed in ventilated H1N1 patients, where mortality was significantly higher (χ^2^ = 35.11, *p* ≤ 0.001, 95% CI [33.97, 36.18]; Figure 4C). In this subgroup, elevated APACHE IV (χ^2^ = 9.369, *p* = 0.002, HR = 1.154), SOFA (χ^2^ = 5.743, *p* = 0.017, HR = 2.599), and pulmonary damage severity index (χ^2^ = 4.284, *p* = 0.038, HR = 1.118) were each independently associated with increased mortality, while CCI was not significant (χ^2^ = 0.378, *p* = 0.539).

In COVID-19 patients, higher APACHE VI (HR = 1.03) and pulmonary damage severity index (HR = 1.26) were linked to increased mortality. In H1N1 patients, higher APACHE VI (HR = 1.15), SOFA (HR = 2.6), and pulmonary damage severity index (HR = 1.12) were associated with higher mortality.

### 3.3. Acute Respiratory Distress Syndrome, Severe and Critical Pneumonia, and Severity of Lung Damage

To investigate clinical severity, patients were classified according to the presence and extent of acute respiratory distress syndrome (ARDS), as well as the type of pneumonia (severe vs. critical). Among the total cohort, 50 patients did not meet ARDS criteria, with the majority being COVID-19 cases (*n* = 38, 76%; *p* ≤ 0.001). In contrast, 98 patients presented with ARDS: 34 were classified as mild (29 COVID-19, 85.3%; *p* ≤ 0.001), 30 as moderate (21 COVID-19, 70%; *p* = 0.045), and 34 as severe (10 COVID-19, 29.4%; *p* = 0.026).

Regarding pneumonia classification, 94 patients were diagnosed with severe pneumonia, of whom 74 had COVID-19 (78.7%; *p* ≤ 0.001). The remaining 54 patients were categorized as having critical pneumonia, and although 30 of these were H1N1 cases (55.56%), the difference in distribution between viral etiologies was not statistically significant (*p* = 0.541).

Comparison of ARDS severity across viral groups revealed that patients with H1N1 were significantly more likely to develop severe ARDS (χ^2^ = 4.080, *p* ≤ 0.001, 95% CI [3.92, 4.17]), whereas patients with COVID-19 were more commonly observed with mild or moderate ARDS or no ARDS at all. Additionally, H1N1 patients were more frequently affected by critical pneumonia (χ^2^ = 4.230, *p* ≤ 0.001, 95% CI [4.16, 4.33]), whereas severe pneumonia predominated in COVID-19 cases.

Pulmonary damage was further stratified by severity using the pulmonary damage severity index. Lung involvement was categorized as mild (index 1–15), moderate (16–22), or severe (23–25). The distribution of damage severity did not significantly differ between groups (χ^2^ = −1.483, *p* = 0.138). COVID-19 patients had a median severity index score of 18 (IQR: 13.75–23.25), while the H1N1 group had a median of 17 (IQR: 11.5–22), with no statistically significant difference between medians (*p* = 0.174).

In this study, 60 patients were classified with mild lung involvement (36 COVID-19, 60%; *p* = 0.156), 49 with moderate damage (33 COVID-19, 67.35%; *p* = 0.022), and 39 with severe lung damage (29 COVID-19, 74.36%; *p* = 0.004), based on CT imaging reviewed at the time of admission.

Mild and moderate ARDS were more common in COVID-19 patients, whereas severe ARDS was more frequent in H1N1 patients. H1N1 patients were more likely to develop severe ARDS, while COVID-19 patients were more prone to critical pneumonia.

## 4. Discussion

While RT-PCR testing remains the gold standard for diagnosing SARS-CoV-2 infections due to its high specificity, chest CT has demonstrated superior sensitivity in detecting pulmonary involvement, particularly in symptomatic or critically ill patients [17]. To our knowledge, this study represents one of the first prospective comparisons of CT-based pulmonary findings, mechanical ventilation requirements, ARDS severity, and 30-day mortality in patients with RT-PCR-confirmed COVID-19 and influenza A H1N1 pneumonia.

As the COVID-19 pandemic progressed, SARS-CoV-2 evolved into variants with lower virulence, notably Omicron, which was associated with a markedly reduced incidence of pneumonia and mortality. A recent study reported that only 53% of patients who died with COVID-19 during the Omicron wave had radiologic evidence of pneumonia, compared to 83% during the ancestral strain period—an absolute reduction of 30% [18]. Simultaneously, there was a notable reduction in seasonal influenza activity worldwide. In mainland China, for instance, influenza-positive rates dropped sharply after the implementation of COVID-19 control measures, and the flu season ended much earlier than in previous years [19]. These trends coincided with widespread adoption of nonpharmaceutical interventions (NPIs), including mask use, physical distancing, hand hygiene, and reduced mobility. Evidence from population-level models indicates that widespread mask use can delay the onset and reduce the transmission of influenza during a pandemic [20]. It has been reported that mask use can help delay an influenza epidemic, but the effectiveness depends on viral transmission characteristics, mask filtration efficiency, and population-level adherence. The observed decline in H1N1 pneumonia cases may thus reflect both the implementation of these NPIs and increased diagnostic attention toward COVID-19. These measures likely disrupted the typical circulation patterns of influenza viruses, causing flu seasons to end earlier and at lower peaks than usual. Consequently, our findings should be interpreted within the context of early pandemic waves dominated by more virulent SARS-CoV-2 variants and strong public health responses that may have had collateral benefits on influenza control.

Imaging in COVID-19 patients typically showed ground-glass opacities and consolidation, whereas pleural effusion was more frequently detected in individuals with H1N1. Notably, COVID-19 imaging more often revealed vascular thickening, crazy paving, and peripheral or lower lobe ground-glass opacities, consistent with prior reports [21]. In contrast, features such as bronchiectasis and more frequent pleural effusion aligned more closely with the H1N1 profile.

Consistent with other studies, SARS-CoV-2-related pneumonia demonstrated a tendency toward bilateral and peripheral lung involvement, especially in the lower lobes [10,11,12]. However, some reviews have suggested overlapping characteristics between the two diseases, with peripheral or peribronchovascular ground-glass opacities, consolidation, subpleural lines, and bronchial distortion reported in both conditions [21].

Historically, severe influenza A (H1N1) infection has been associated with high mortality in critical care settings—up to 41% in some cohorts [22]. In our study, patients with H1N1 exhibited a higher likelihood of requiring mechanical ventilation, supported by higher SOFA and APACHE IV scores. The regression model showed that patients with elevated SOFA scores had a 6.5-fold greater risk of requiring mechanical ventilation, while each point increase in APACHE IV increased the likelihood by 1.3 times. Conversely, a lower comorbidity burden (as measured by CCI) was associated with reduced ventilation needs.

When evaluating mortality, patients with higher pulmonary damage severity indices experienced worse outcomes, regardless of viral etiology. Among COVID-19 cases, elevated APACHE IV and pulmonary damage scores were associated with increased 30-day mortality (HR = 1.03 and HR = 1.26, respectively). In H1N1 patients, additional risk factors included elevated SOFA (HR = 2.6), APACHE IV (HR = 1.15), and pulmonary damage severity index (HR = 1.12). The difference in SOFA scores likely reflects the distinct pathophysiologic profiles of the two infections. H1N1 is more commonly associated with multiorgan dysfunction, contributing to higher SOFA scores, whereas COVID-19 tends to cause more isolated pulmonary injury. This distinction underscores the need for virus-specific triage and treatment strategies in critical care. The apparent discrepancy in survival rates—70% overall vs. ~20% when stratified by etiology—reflects the unequal distribution of outcomes across viral subgroups. COVID-19 patients requiring ventilation experienced significantly higher mortality compared to H1N1 patients, which reduced the overall survival when groups were separated. This highlights the importance of etiologic context when interpreting aggregated survival data.

Differences in ARDS severity were also evident between groups. Mild and moderate ARDS were more prevalent in COVID-19 patients, while severe ARDS occurred more frequently in those with H1N1. Notably, severe pneumonia was more commonly observed in COVID-19, while the distribution of critical pneumonia did not differ significantly between the two groups. These findings are consistent with prior literature indicating that underlying conditions, including diabetes and chronic obstructive pulmonary disease (COPD), are predictive of disease progression and respiratory failure in viral pneumonias [23]. These radiologic and clinical differences may also be attributed to divergent infection pathways. SARS-CoV-2 primarily targets ACE2-expressing type II alveolar epithelial cells, resulting in peripheral alveolar damage. In contrast, H1N1 typically infects the upper respiratory tract, leading to a different pattern of clinical and imaging findings.

Our findings align with previous studies showing that COVID-19 is often characterized by extensive ground-glass opacities and peripheral distribution, while H1N1 tends to present with pleural effusion and consolidation. These radiologic patterns, combined with clinical scoring systems, can support early triage decisions—such as prioritizing ICU-level care for H1N1 patients with high SOFA scores or initiating aggressive pulmonary monitoring in COVID-19 cases with high APACHE IV scores. Identifying these distinct profiles is clinically meaningful for tailoring treatment plans, allocating resources, and optimizing outcomes, especially during periods of overlapping viral circulation.

Finally, although both diseases shared radiologic and clinical overlap, the prognostic trajectories diverged. H1N1 was more frequently associated with severe ARDS and invasive ventilation, while COVID-19 cases often presented with extensive lung damage and a higher prevalence of radiologic abnormalities. Our findings highlight how early CT imaging and severity scoring systems can guide clinical decision-making in patients with viral pneumonia.

### 4.1. Limitations

This study has several limitations that should be considered when interpreting the findings. First, follow-up chest CT scans were not available, limiting our ability to track the progression or resolution of pulmonary lesions, including fibrosis. Second, some patients had initiated antibiotic or corticosteroid therapy prior to hospital admission, which may have influenced the initial imaging features observed upon evaluation.

Third, the study did not investigate the potential role of coinfections, such as concurrent bacterial pneumonia, which could have confounded both imaging findings and clinical outcomes. Histopathological confirmation through lung biopsies or autopsies was not performed, which restricts the ability to directly correlate tomographic abnormalities with underlying tissue pathology.

This study did not include follow-up imaging, which restricts our ability to assess the evolution or resolution of pulmonary findings over time, including potential fibrosis. Furthermore, while corticosteroid therapy was commonly used as part of clinical care, we did not perform a systematic analysis of treatment effects or dosage timing due to variability in clinical documentation. These aspects represent important areas for future prospective studies.

Additionally, although the overall sample size was sufficient for the primary analyses, some subgroup comparisons—such as predictors of mortality in mechanically ventilated patients—may have been underpowered, potentially limiting the detection of smaller but clinically meaningful effects.

### 4.2. Conclusions

In this comparative study of patients with RT-PCR-confirmed COVID-19 and influenza A H1N1 pneumonia, distinct radiologic and clinical patterns were observed between the two groups. Ground-glass opacities and consolidation were more frequently detected in COVID-19 cases, while pleural effusion was significantly more common among patients with H1N1. COVID-19 patients were also more likely to exhibit bilateral and peripherally distributed lesions on CT imaging.

In terms of respiratory support, patients with H1N1 demonstrated a greater need for invasive mechanical ventilation, with elevated SOFA and APACHE IV scores emerging as significant predictors. Conversely, a lower Charlson Comorbidity Index was associated with reduced ventilation risk. The severity of lung involvement, as measured by the pulmonary damage severity index, was closely linked to 30-day mortality outcomes in both groups.

Among COVID-19 patients, elevated APACHE IV scores and higher pulmonary damage severity were associated with increased mortality risk. In H1N1 patients, mortality was independently associated with higher APACHE IV, SOFA, and pulmonary damage scores. Although mild and moderate ARDS were more prevalent in the COVID-19 group, severe ARDS was significantly more common in H1N1 cases. COVID-19 was also more frequently associated with severe pneumonia, while the incidence of critical pneumonia was comparable between groups. These findings emphasize the importance of CT imaging and clinical severity scoring systems in distinguishing between viral pneumonias and in guiding timely and tailored management strategies.

## Figures and Tables

**Figure 1 diagnostics-15-01430-f001:**
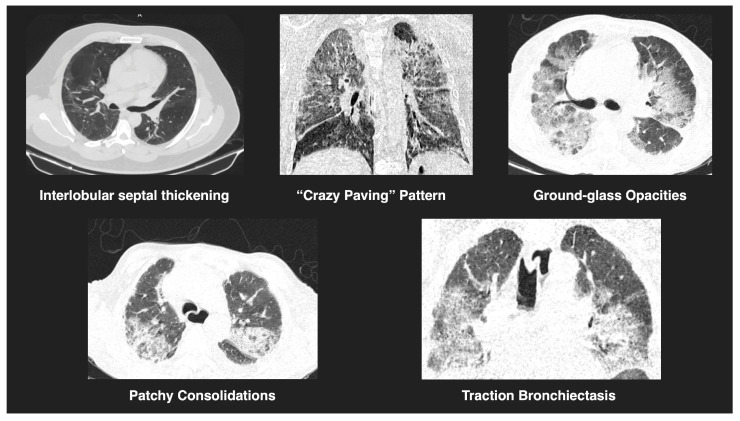
Representative tomographic features of viral pneumonia. CT images show key findings observed in COVID-19 and H1N1 cases, including interlobular septal thickening, a “crazy paving” pattern, ground-glass opacities, patchy consolidations, and traction bronchiectasis. These features reflect varying degrees of pulmonary involvement and are consistent with viral pneumonitis.

**Figure 2 diagnostics-15-01430-f002:**
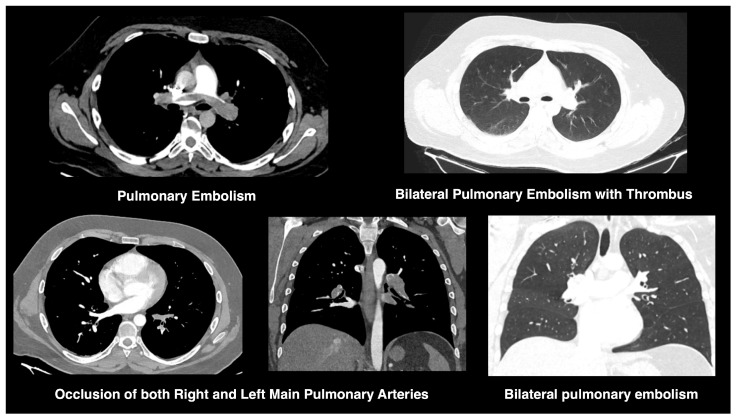
CT findings of pulmonary embolism in COVID-19. CT angiograms demonstrate bilateral pulmonary embolism, with thrombus visualized in both main pulmonary arteries. Findings include vascular occlusion and associated parenchymal ground-glass opacities, compatible with COVID-19-related embolic events.

**Figure 3 diagnostics-15-01430-f003:**
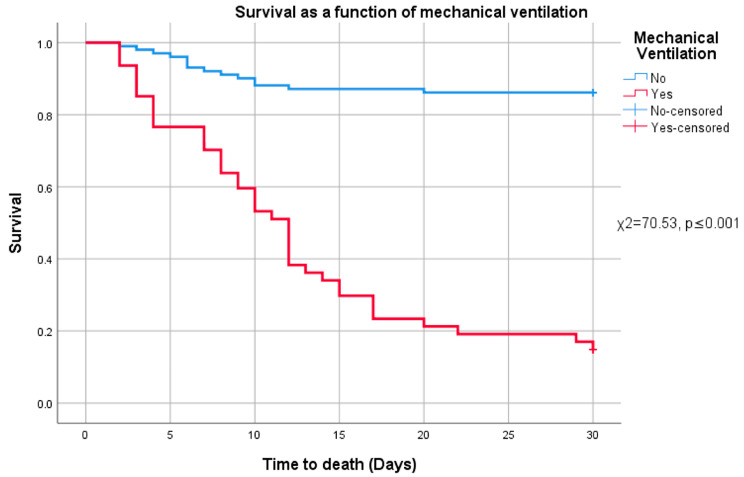
Survival and mechanical ventilation.

**Figure 4 diagnostics-15-01430-f004:**
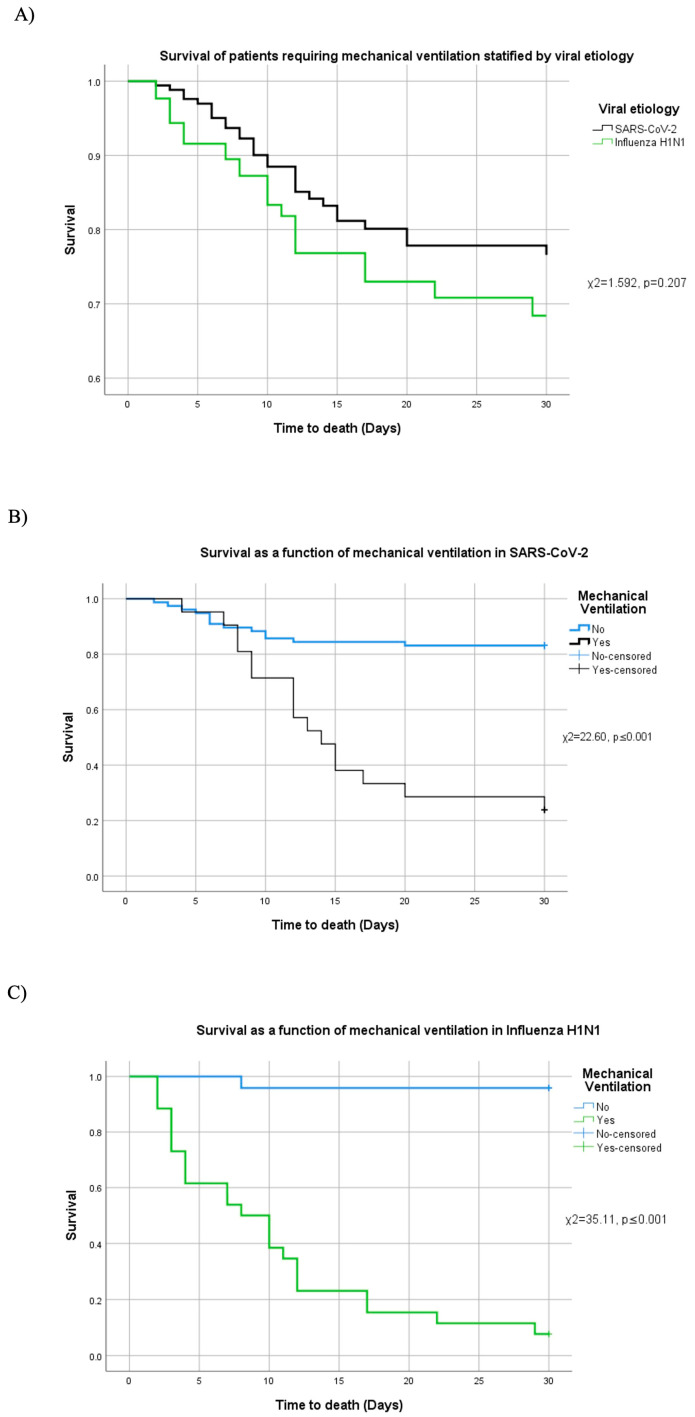
Survival and mechanical ventilation stratified by etiology. (**A**) Survival among patients who required mechanical ventilation by viral etiology. Patient survival comparing the need (or not) of mechanical ventilation in (**B**) SARS-CoV-2 and (**C**) H1N1.

**Table 1 diagnostics-15-01430-t001:** Distribution of clinical and imaging variables in COVID-19 and H1N1.

Variable	Population	COVID-19	H1N1	Percentage (COVID-19)	*p*-Value
Clinical findings					
Total patients	148	98	50	66.2	-
Age	56.7 (±14.75)	58.3 (±14.98)	55.5 (±14.36)	-	0.490
Body mass index	28.2 (±2.87)	28.3 (±3.08)	28 (±2.41)	-	0.520
Acute Physiology and Chronic Health Evaluation	22 *	20 *	36 *	-	0.001
Sequential Organ Failure Assessment	2 *	2 *	5 *	-	≤0.001
Charlson Comorbidity Index	1 *	1 *	1.5 *	-	0.707
Pulmonary damage severity index	18 *	18 *	17 *	-	0.178
Radiological findings					
Consolidation	130	80	50	61.5	0.001
Ground-glass opacities	106	95	11	89.6	≤0.001
Vascular thickening	95	66	29	69.5	0.264
Lymphatic ganglia	71	42	29	59.2	0.082
Crazy paving pattern	42	32	10	76.2	0.108
Pleural effusion	26	6	20	13.1	≤0.001
Atelectasis	17	15	2	88.2	0.042
Tree-in-bud pattern	8	6	2	75	0.590
Embolism	9	7	2	77.8	0.183
Bilateral lesions	133	95	38	96.9	≤0.001
Peripheral lesions	122	95	27	77.9	≤0.001
Basal lobule lesions	141	95	46	67.4	0.182

* Reported as median. COVID-19: SARS-CoV-2 infection. H1N1: Influenza A H1N1 infection.

**Table 2 diagnostics-15-01430-t002:** Logistic regression model coefficients and effect sizes.

Mechanical Ventilation (*n* = 47)									
				95% CI for Odds Ratio			Pseudo-R^2^		
Variable	χ^2^	*p*	b	Lower	Odds	Upper	H&L	C&S	Negelkerke
APACHE IV	26.996	≤0.001	0.256	1.173	1.292	1.423	0.303	0.378	0.530
SOFA	9.920	0.002	1.866	2.024	6.464	20.650	0.145	0.056	0.079
APACHE IV × SOFA	8.876	0.003	−0.033	0.947	0.968	0.989	0.144	0.019	0.069
APACHE IV × CCI	8.313	0.004	−0.036	0.941	0.964	0.988	0.072	0.026	0.036
Charlson Comorbidity Index	5.996	0.014	1.203	0.510	0.673	0.889	0.089	0.031	0.043
Pulmonary damage severity index	1.838	0.175	0.073	0.968	1.076	1.196	0.001	0.007	0.010

CI: confidence interval. χ^2^: Wald test. H&L: Hosmer and Lemeshow R^2^. C&S: Cox and Snell R^2^. APACHE IV: Acute Physiology and Chronic Health Evaluation. SOFA: Sequential Organ Failure Assessment. CCI: Charlson Comorbidity Index. Beta value refers to the measure of the modeled effect that reflects the parameter estimate. All reported *p*-values are corrected with a Bonferroni correction for multiple comparisons. “×” denotes interaction term.

## Data Availability

The data presented in this study are available on request from the corresponding author. The data are not publicly available due to sensitive clinical data that cannot be fully anonymized.

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
