# Peer review of "Comparison of Tomographic Findings in Patients with Influenza A (H1N1) and COVID-19"

_diagnostics, 2025, doi:10.3390/diagnostics15111430_

Round 1

Reviewer 1 Report

Comments and Suggestions for Authors

Thank you for the opportunity to review the work titled "Comparison of Tomographic Findings in Patients with Influenza A (H1N1) and COVID-19." The article is well-written, but I have some suggestions for improvement:

- Authors should specify the waves of the COVID-19 pandemic, as pneumonia was less frequent during the Omicron waves.

- Authors should specify how many COVID-19 patients and influenza A patients showed lung embolism.

- I suggest deleting the word "summary" (line 282) at the end of the results.

- I recommend dividing FIG 1 and adding images that represent the distribution of COVID-19 pneumonia and influenza A pneumonia. Summarizing the figure legends of FIG 1 would also be beneficial. Additionally, creating FIG 2 to represent lung thromboembolism in COVID-19 pneumonia would be valuable.

- In the discussion, I suggest adding a sentence about the evolution of the COVID-19 pandemic, from more aggressive variants causing pneumonia to the Omicron variants that less frequently cause pneumonia. It may be worth exploring if the use of masks played a role, as the circulation of the Influenza virus was less during the pandemic. Authors should share their opinion in the context of the existing literature.

Author Response

Response to the reviewers

Dear Editors,

We appreciate the opportunity to revise and resubmit our manuscript titled “Comparison of Tomographic Findings in Patients with Influenza A (H1N1) and COVID-19” (Manuscript ID: 3586583). We thank the reviewers for their thoughtful and constructive feedback. We have carefully addressed each comment and revised the manuscript accordingly to improve its clarity, rigor, and overall contribution. Below we provide a point-by-point response to the reviewers’ comments.

Comment: “Authors should specify the waves of the COVID-19 pandemic, as pneumonia was less frequent during the Omicron waves.”

Response: We appreciate this insightful suggestion. Our study includes cases from June 2020 to May 2021, corresponding to the first and second waves of the COVID-19 pandemic in Mexico, when ancestral and pre-Delta variants of SARS-CoV-2 predominated. We acknowledge that pneumonia has become less frequent during later waves, particularly those dominated by Omicron variants, as noted in recent literature. However, our findings reflect clinical and radiological features primarily associated with earlier, more virulent strains, and thus are most applicable to those phases of the pandemic.

Text added: “… a period corresponding to the first and second waves of the COVID-19 pandemic in Mexico, during which ancestral variants of SARS-CoV-2 predominated prior to the emergence of Omicron-related strains.” (Page 3, lines 117-119)

Comment: “Authors should specify how many COVID-19 patients and influenza A patients showed lung embolism.”

Response: Thank you for this observation. We have clarified in the Results section that a total of nine patients were diagnosed with pulmonary embolism: seven with COVID-19 and two with H1N1.

Text added: “Pulmonary embolism was identified in a total of nine patients, including seven with COVID-19 and two with H1N1. The difference between groups was not statistically significant (p = 0.183).” (Page 8, lines 318-320)

Comment: “I suggest deleting the word “summary” (line 282) at the end of the results.”

Response: The word “summary” has been removed as suggested at the end of each relevant Results subsection.

Comment: “I recommend dividing FIG 1 and adding images that represent the distribution of COVID-19 pneumonia and influenza A pneumonia. Summarizing the figure legends of FIG 1 would also be beneficial. Additionally, creating FIG 2 to represent lung thromboembolism in COVID-19 pneumonia would be valuable.”

Response: We have divided the original figure into two separate figures:

  • Figure 1: Focuses on representative CT findings in COVID-19 and H1N1 pneumonia.
  • Figure 2: Illustrates CT evidence of pulmonary embolism in COVID-19 patients. Both figure legends have been abridged and clarified.

Text added:

Figure legends

Figure 1. Representative tomographic features of viral pneumonia.

CT images show key findings observed in COVID-19 and H1N1 cases, including interlobular septal thickening, “crazy paving” pattern, ground-glass opacities, patchy consolidations, and traction bronchiectasis. These features reflect varying degrees of pulmonary involvement and are consistent with viral pneumonitis.

Figure 2. CT findings of pulmonary embolism in COVID-19.

CT angiograms demonstrate bilateral pulmonary embolism, with thrombus visualized in both main pulmonary arteries. Findings include vascular occlusion and associated parenchymal ground-glass opacities, compatible with COVID-19-related embolic events.

Comment: “In the discussion, I suggest adding a sentence about the evolution of the COVID-19 pandemic, from more aggressive variants causing pneumonia to the Omicron variants that less frequently cause pneumonia. It may be worth exploring if the use of masks played a role, as the circulation of the Influenza virus was less during the pandemic. Authors should share their opinion in the context of the existing literature.”

Response: Thank you for the thoughtful recommendation. We have expanded the Discussion section to address the transition from more virulent early variants to the less severe Omicron variant, supported by recent evidence. Specifically, we now include recent evidence showing that the incidence of COVID-19 pneumonia decreased significantly during the Omicron era compared to the ancestral strain. A recent study (Hammer et al., Acad Radiol, 2024) reported a 30% absolute reduction in the prevalence of pneumonia among patients who died with COVID-19 in the Omicron era versus the ancestral era, highlighting the reduced virulence of later variants. In addition, we comment on how public health interventions—such as widespread mask use and social distancing—likely contributed to the suppressed circulation of seasonal influenza viruses during the same timeframe. These additions provide context for interpreting our findings, which are more relevant to the early, more aggressive variants circulating during the first pandemic waves.

Text added: “As the COVID-19 pandemic progressed, SARS-CoV-2 evolved into variants with lower virulence, notably Omicron, which was associated with a markedly reduced incidence of pneumonia and mortality. A recent study reported that only 53% of patients who died with COVID-19 during the Omicron wave had radiologic evidence of pneumonia, compared to 83% during the ancestral strain period—an absolute reduction of 30%.20 Simultaneously, there was a notable reduction in seasonal influenza activity worldwide. In mainland China, for instance, influenza-positive rates dropped sharply after the implementation of COVID-19 control measures, and the flu season ended much earlier than in previous years.21 These trends coincided with widespread adoption of nonpharmaceutical interventions (NPIs), including mask use, physical distancing, hand hygiene, and reduced mobility. Evidence from population-level models indicates that widespread mask use can delay the onset and reduce the transmission of influenza during a pandemic.22 It has been reported that mask use can help delay an influenza epidemic, but the effectiveness depends on viral transmission characteristics, mask filtration efficiency, and population-level adherence. The observed decline in H1N1 pneumonia cases may thus reflect both the implementation of these NPIs and increased diagnostic attention toward COVID-19. These measures likely disrupted the typical circulation patterns of influenza viruses, causing flu seasons to end earlier and at lower peaks than usual. Consequently, our findings should be interpreted within the context of early pandemic waves dominated by more virulent SARS-CoV-2 variants and strong public health responses that may have had collateral benefits on influenza control.” (Page 14, lines 465-483)

References added:

  1. Hammer MM, Sodickson AD, Marshall AD, Faust JS. Prevalence of Pneumonia Among Patients Who Died with COVID-19 Infection in Ancestral Versus Omicron Variant Eras. Acad Radiol. 2024;31(1):1-6. doi:10.1016/j.acra.2023.05.008. PMID: 37271637
  2. Liu M, Deng L, Wang D, Jiang T. Influenza activity during the outbreak of coronavirus disease 2019 in Chinese mainland. Biosaf Health. 2020;2(4):206-209. doi:10.1016/j.bsheal.2020.08.005. PMID: 32905055
  3. Brienen NC, Timen A, Wallinga J, van Steenbergen JE, Teunis PF. The effect of mask use on the spread of influenza during a pandemic. Risk Anal. 2010;30(8):1210-1218. doi:10.1111/j.1539-6924.2010.01428.x. PMID: 20497389

We thank the reviewers for their constructive feedback, which has greatly improved the clarity and quality of our manuscript. We have addressed each comment thoroughly and incorporated the suggested changes. We hope the revised version meets the expectations of the reviewers and editorial team, and we remain available for further revisions if necessary.

Sincerely,

Jaime D. Mondragon, M.D., Ph.D.

University Medical Center Groningen

San Diego State University

Universidad Nacional Autónoma de México

Reviewer 2 Report

Comments and Suggestions for Authors

The authors have made an interesting prospective study concerning a comparison of pulmonary CT findings in SARS-CoV-2 and influenza A H1N1 in order to differentiate the imaging characteristics between these two viral infections by analyzing demographic profiles, tomographic features, and prognostic markers. The authors also tried to assess the likelihood of requiring invasive mechanical ventilation and to evaluate whether higher disease severity correlates with increased mortality.

The study is still clinically important. It demonstrated that patients with H1N1 were more likely (HR=1.16) to require mechanical ventilation compared to those with COVID-19. Higher SOFA and APACHE IV scores were associated with a 6.5-fold and 1.3-fold increased likelihood, respectively, of requiring invasive mechanical ventilation. In COVID-19 patients, higher APACHE VI (HR=1.03) and pulmonary damage severity index (HR=1.26) were linked to increased mortality. In H1N1 patients, higher APACHE VI (HR=1.15), SOFA (HR=2.6), and pulmonary damage severity index (HR=1.12) were associated with higher mortality. Furthermore, the study shows that mild and moderate ARDS were more common in COVID-19 patients, whereas severe ARDS was more frequent in H1N1 patients. H1N1 patients were more likely to develop severe ARDS, while COVID-19 patients were more prone to critical pneumonia.

Al in all, interesting and important subject, radiologically known already results, but analysis of these rdiological findings with clinical scales is interesting. The biggest drawback is the lack of follow-up examinations and anylysis with the cortical treatment. Good introduction, description of methods, tables, figures, limitations, results and discussion. Good English.

I haven’t any major clinical drawbacks.

I therefore suggest to accept this article .

Author Response

Dear Editors,

We appreciate the opportunity to revise and resubmit our manuscript titled “Comparison of Tomographic Findings in Patients with Influenza A (H1N1) and COVID-19” (Manuscript ID: 3586583). We thank the reviewers for their thoughtful and constructive feedback. We have carefully addressed each comment and revised the manuscript accordingly to improve its clarity, rigor, and overall contribution. Below we provide a point-by-point response to the reviewers’ comments.

We thank Reviewer 2 for their positive evaluation of our study’s clinical relevance and methodological clarity.

Comment: “The biggest drawback is the lack of follow-up examinations and analysis with the cortical treatment.”

Response: We acknowledge that follow-up CT scans were not available, limiting our ability to assess longitudinal changes such as fibrosis. Additionally, although corticosteroid therapy was part of standard management, a systematic analysis of its effects was not feasible. These limitations are now noted in the Discussion.

Text added: “This study did not include follow-up imaging, which restricts our ability to assess the evolution or resolution of pulmonary findings over time, including potential fibrosis. Furthermore, while corticosteroid therapy was commonly used as part of clinical care, we did not perform a systematic analysis of treatment effects or dosage timing due to variability in clinical documentation. These aspects represent important areas for future prospective studies.” (Page 15, lines 545-550)

We thank the reviewers for their constructive feedback, which has greatly improved the clarity and quality of our manuscript. We have addressed each comment thoroughly and incorporated the suggested changes. We hope the revised version meets the expectations of the reviewers and editorial team, and we remain available for further revisions if necessary.

Sincerely,

Jaime D. Mondragon, M.D., Ph.D.

University Medical Center Groningen

San Diego State University

Universidad Nacional Autónoma de México

Reviewer 3 Report

Comments and Suggestions for Authors

Comment

This paper presents a radiological and clinical analysis of patients with COVID-19 and influenza A H1N1 pneumonia.

The difference between the CT images of COV2 patients and H1N1 patients is linked to the difference in clinical course. It was also shown that the ratio of mild to severe patients differs between the two groups. This is likely to be very useful for doctors treating patients as a guide for treatment.

The high rate of invasive mechanical ventilation among H1N1 patients and the indication of SOFA and APACHE IV scores as prognostic factors provide guidelines for determining whether intensive care is appropriate, especially for the elderly.

In COVID-19 patients, APACHE IV scores and higher pulmonary damage severity were associated with death.

Interestingly, SOFA scores are missing. This is thought to indicate an important factor in the difference in the infection patterns between H1N1 and COVID-19.

If possible, it would be good to mention this difference in the discussion.

COVID-19 infects type II alveolar epithelial cells, which contain ACE2, and damages peripheral alveoli. On the other hand, H1N1virus is characterized by upper respiratory tract infection. This infection pattern may represent a distinctive feature of the two infectious diseases.

Miner Comment

1)There is no SD value in the patient data in Table 1.

2)There are no Figure 3s.

3) In Figure 2 (survival by etiology and by ventilation), the combined survival rate for both groups of patients who required ventilation was approximately 70%. However, when divided based on viral etiology, the survival rate for patients who required ventilation was approximately 20%.

What is the difference?

Other figures for survival rates for patients on ventilators, broken down by the type of virus they were infected with, were also around 20%

Author Response

Dear Editors,

We appreciate the opportunity to revise and resubmit our manuscript titled “Comparison of Tomographic Findings in Patients with Influenza A (H1N1) and COVID-19” (Manuscript ID: 3586583). We thank the reviewers for their thoughtful and constructive feedback. We have carefully addressed each comment and revised the manuscript accordingly to improve its clarity, rigor, and overall contribution. Below we provide a point-by-point response to the reviewers’ comments.

Comment: “Interestingly, SOFA scores are missing. This is thought to indicate an important factor in the difference in the infection patterns between H1N1 and COVID-19. If possible, it would be good to mention this difference in the discussion.”

Response: Thank you for this observation. SOFA scores were included in the Discussion section and are summarized in Table 1. We now emphasize the relevance of SOFA in differentiating the infection profiles of H1N1 and COVID-19 in the Discussion.

Text Added: “The difference in SOFA scores likely reflects the distinct pathophysiologic profiles of the two infections. H1N1 is more commonly associated with multiorgan dysfunction, contributing to higher SOFA scores, whereas COVID-19 tends to cause more isolated pulmonary injury. This distinction underscores the need for virus-specific triage and treatment strategies in critical care.” (Page 14-15, lines 507–511)

 Comment: “COVID-19 infects type II alveolar epithelial cells, which contain ACE2, and damages peripheral alveoli. On the other hand, H1N1 virus is characterized by upper respiratory tract infection. This infection pattern may represent a distinctive feature of the two infectious diseases.

Response: We appreciate this suggestion and have incorporated a brief mechanistic comparison into the Discussion to support our radiological and clinical findings.

Text added: “These radiologic and clinical differences may also be attributed to divergent infection pathways. SARS-CoV-2 primarily targets ACE2-expressing type II alveolar epithelial cells, resulting in peripheral alveolar damage. In contrast, H1N1 typically infects the upper respiratory tract, leading to a different pattern of clinical and imaging findings.” (Page 15, lines 519–522)

Minor Comment

Comment: “There is no SD value in the patient data in Table 1.”

Response: Thank you for pointing this out. The SD values for continuous variables (e.g., age, BMI) have been added to Table 1 in the revised manuscript.

Comment: There are no Figure 3s.

Response: Thank you for identifying this error. We have corrected the figure numbering and ensured consistency throughout the text and legends.

Comment: “In Figure 2 (survival by etiology and by ventilation), the combined survival rate for both groups of patients who required ventilation was approximately 70%. However, when divided based on viral etiology, the survival rate for patients who required ventilation was approximately 20%. What is the difference?” Other figures for survival rates for patients on ventilators, broken down by the type of virus they were infected with, were also around 20%

Response: We appreciate this important question. The apparent discrepancy arises from how survival is stratified. The 70% survival rate reflects the overall outcome for all ventilated patients combined. However, when subdivided by etiology, survival differences are more pronounced, especially among COVID-19 patients, who experienced higher mortality. We have clarified this in the Results and added an explanatory note to the Discussion.

Text added: “The apparent discrepancy in survival rates—70% overall vs. ~20% when stratified by etiology—reflects the unequal distribution of outcomes across viral subgroups. COVID-19 patients requiring ventilation experienced significantly higher mortality compared to H1N1 patients, which reduced the overall survival when groups were separated. This highlights the importance of etiologic context when interpreting aggregated survival data.” (Page 15, lines 511–516)

We thank the reviewers for their constructive feedback, which has greatly improved the clarity and quality of our manuscript. We have addressed each comment thoroughly and incorporated the suggested changes. We hope the revised version meets the expectations of the reviewers and editorial team, and we remain available for further revisions if necessary.

Sincerely,

Jaime D. Mondragon, M.D., Ph.D.

University Medical Center Groningen

San Diego State University

Universidad Nacional Autónoma de México

Reviewer 4 Report

Comments and Suggestions for Authors

Dear Authors,

This manuscript presents a prospective comparison of tomographic findings in patients with severe and critical pneumonia due to either SARS-CoV-2 or influenza A H1N1. The study is timely and addresses an important clinical question regarding the differentiation of imaging patterns in two significant viral respiratory infections. However, several areas of the manuscript require improvement to enhance its clarity, rigor, and overall scientific value.

First, the abstract is overly long and contains excessive background detail that would be better suited for the introduction; a more concise summary of objectives, methods, key results, and conclusions would improve its readability.

Second, there are notable issues with language and grammar throughout the manuscript, including awkward phrasing, inconsistent terminology, and excessive passive voice, which may hinder comprehension for an international audience. Careful language editing by a native English speaker or professional editor is strongly recommended.

Third, while the methodological description is generally adequate, greater detail is needed regarding the scoring and interpretation of tomographic findings, including whether standardized CT scoring systems (such as CO-RADS or RALE) were employed. Furthermore, the authors mention several prognostic indices (e.g., APACHE IV, SOFA, and CCI), but their application and interpretation could be explained more clearly, particularly in relation to the outcomes of mechanical ventilation and mortality.

Statistical methods are mostly appropriate; however, confidence intervals should be provided for all key effect sizes (e.g., hazard ratios), and the rationale for using certain cutoffs or interaction terms in regression analyses should be justified.

The sample size, although modest, is acceptable; nonetheless, some findings may be underpowered, and this limitation should be acknowledged. The results are interesting, especially the observed differences in lesion distribution and prevalence of pleural effusion between groups.

 The discussion section should more critically engage with the existing literature and elaborate on the clinical implications of the findings, particularly regarding triage and treatment decisions. Additionally, while the study is described as prospective, more information on patient enrollment and data handling is needed to confirm this designation. The conclusion is appropriate but should more clearly highlight the novel contributions of this study.

Finally, minor inconsistencies in author affiliations, citation formatting, and typographic elements (such as duplicated words or abrupt line breaks) should be corrected before publication.

Sincerely

Comments on the Quality of English Language

English needs improvement.

Author Response

Dear Editors,

We appreciate the opportunity to revise and resubmit our manuscript titled “Comparison of Tomographic Findings in Patients with Influenza A (H1N1) and COVID-19” (Manuscript ID: 3586583). We thank the reviewers for their thoughtful and constructive feedback. We have carefully addressed each comment and revised the manuscript accordingly to improve its clarity, rigor, and overall contribution. Below we provide a point-by-point response to the reviewers’ comments.

Comment: “First, the abstract is overly long and contains excessive background detail that would be better suited for the introduction; a more concise summary of objectives, methods, key results, and conclusions would improve its readability.”

Response: We appreciate this feedback. The abstract has been revised to be more concise and now focuses on the primary objectives, study design, key findings, and conclusions. Background detail has been moved to the Introduction section.

Text added: “Objectives: To identify CT-based predictors of mechanical ventilation and mortality in patients with severe and critical viral pneumonia, and to examine the association between imaging severity and outcomes in ventilated patients.

Methods: We analyzed pulmonary CT scans from 148 patients with severe or critical pneumonia caused by COVID-19 (n=98) or influenza A H1N1 (n=50). Patients were assessed based on tomographic patterns, demographics, clinical severity scores (Charlson Comorbidity Index, SOFA, APACHE IV), and biomarkers. Survival analyses were performed using Kaplan-Meier curves and multivariable Cox regression.

Results: Bilateral, peripheral, and basal lung involvement was common across both groups. Ground-glass opacities (89.62%, p ≤ 0.001) and consolidation (61.54%, p = 0.001) were more prevalent in COVID-19, whereas pleural effusion was significantly more frequent in H1N1 (76.92%, p ≤ 0.001). COVID-19 cases more often presented with bilateral (96.94%) and peripheral lesions (77.87%). H1N1 patients were more likely to develop severe ARDS and require mechanical ventilation. In COVID-19, higher APACHE IV scores and pulmonary damage severity index were independently associated with increased mortality.

Conclusions: Radiologic and clinical severity profiles differ between COVID-19 and H1N1 pneumonia. CT-based assessments combined with prognostic scores may aid early risk stratification and guide treatment decisions in patients with severe viral pneumonia.”

Comment: “Second, there are notable issues with language and grammar throughout the manuscript, including awkward phrasing, inconsistent terminology, and excessive passive voice, which may hinder comprehension for an international audience. Careful language editing by a native English speaker or professional editor is strongly recommended.”

Response: We appreciate this observation. The manuscript has undergone a thorough language review by a native English speaker with expertise in medical writing. We revised sections containing awkward phrasing, standardized terminology (e.g., use of “ground-glass opacities” vs. “ground glass images”), and reduced the use of passive voice to improve clarity and readability. These revisions were made throughout the manuscript to enhance its accessibility for an international audience.

Comment: “Third, while the methodological description is generally adequate, greater detail is needed regarding the scoring and interpretation of tomographic findings, including whether standardized CT scoring systems (such as CO-RADS or RALE) were employed.”

Response: We appreciate the reviewer’s suggestion. In response, we have expanded the Radiological Assessment subsection of the Methods to clarify the scoring approach. Specifically, we now state that the study used a semi-quantitative pulmonary damage severity index based on lobar involvement, rather than standardized scoring systems such as CO-RADS or RALE. This index has been validated in prior studies (e.g., Chang et al., 2005; Mahdjoub et al., 2020) and was chosen for its applicability across viral pneumonias. Appropriate citations have been included to support the methodology.

Text added: “All chest CT scans were assessed for characteristic imaging features (Figure 1), including consolidation, ground-glass opacities, nodular lesions, lymphadenopathy, vascular thickening, and evidence of embolism (Figure 2). To quantify lung involvement, we used a semi-quantitative pulmonary damage severity index based on estimated parenchymal involvement in each of the five lung lobes. Each lobe was scored from 0 (no involvement) to 5 (more than 75% involvement), yielding a total score ranging from 0 to 25 per patient. This approach, previously validated in similar contexts,17,18 was selected over standardized diagnostic scoring systems such as CO-RADS or RALE due to its flexibility and broader applicability in comparative viral pneumonia analysis.” (Pages 4-5, lines 185-195)

Comment: “Furthermore, the authors mention several prognostic indices (e.g., APACHE IV, SOFA, and CCI), but their application and interpretation could be explained more clearly, particularly in relation to the outcomes of mechanical ventilation and mortality.”

Response: We thank the reviewer for highlighting this point. We have revised the Methods and Results sections to clarify the role of each prognostic index. Specifically, we explain that the APACHE IV and SOFA scores were used to quantify acute physiological derangement and organ failure severity upon admission, while the Charlson Comorbidity Index (CCI) provided a measure of baseline comorbidity burden. Each of these indices was included as independent variables in regression models predicting the need for mechanical ventilation and 30-day mortality. The text now explicitly details how each index was applied, how scores were derived, and their statistical relevance to the outcomes examined.

Text added: “Three prognostic scoring systems were used to assess clinical severity and risk: the Charlson Comorbidity Index (CCI), the Sequential Organ Failure Assessment (SOFA), and the Acute Physiology and Chronic Health Evaluation IV (APACHE IV). CCI reflects preexisting comorbidity burden, while SOFA and APACHE IV quantify acute physiological dysfunction and severity of illness at admission. These scores were used as independent variables in logistic regression to predict mechanical ventilation and in Cox regression models to evaluate mortality risk.” (Page 4, 165-171)

“Higher APACHE IV and SOFA scores were significantly associated with increased likelihood of invasive mechanical ventilation. APACHE IV, representing acute physiological burden, and SOFA, indicating multi-organ dysfunction, each independently contributed to risk prediction. In contrast, CCI showed an inverse association with ventilation need, suggesting that lower comorbidity burden may reflect more acute disease severity in hospitalized cases.” (Pages 9-10, lines 357-362)

Comment: “Statistical methods are mostly appropriate; however, confidence intervals should be provided for all key effect sizes (e.g., hazard ratios), and the rationale for using certain cutoffs or interaction terms in regression analyses should be justified.”

Response: Thank you for this important suggestion. We have updated the Results section to ensure that 95% confidence intervals are now reported for all key effect sizes, including odds ratios and hazard ratios derived from logistic and Cox regression models. In the Methods section, we also added a rationale for including interaction terms (e.g., APACHE IV × SOFA), which were tested to explore potential synergistic effects between acute physiological burden and organ failure. A backward stepwise selection approach was used to retain the most informative predictors and their interactions, optimizing model performance based on likelihood ratio tests and pseudo-R² metrics.

Text added: Interaction terms such as APACHE IV × SOFA and APACHE IV × CCI were tested to evaluate whether combined physiological stress and organ dysfunction or comorbidity burden had synergistic effects on outcomes. Backward stepwise elimination was employed to refine the model, using −2 log likelihood and pseudo-R² statistics (Hosmer-Lemeshow, Cox-Snell, and Nagelkerke) to assess fit. (Page 8, lines 288-293).

Comment: “The sample size, although modest, is acceptable; nonetheless, some findings may be underpowered, and this limitation should be acknowledged. The results are interesting, especially the observed differences in lesion distribution and prevalence of pleural effusion between groups.”

Response: We agree with the reviewer’s observation and have added a statement in the Limitations section acknowledging the possibility of underpowered subgroup analyses. While the overall sample size (n=148) was adequate for primary comparisons, we recognize that certain subgroup effects—particularly those related to mortality predictors in ventilated patients—may not have reached statistical significance due to limited power. This has now been explicitly addressed in the manuscript.

Text added: "Additionally, although the overall sample size was sufficient for the primary analyses, some subgroup comparisons—such as predictors of mortality in mechanically ventilated patients—may have been underpowered, potentially limiting the detection of smaller but clinically meaningful effects." (Page 15, lines 551-554)

Comment: “The discussion section should more critically engage with the existing literature and elaborate on the clinical implications of the findings, particularly regarding triage and treatment decisions. Additionally, while the study is described as prospective, more information on patient enrollment and data handling is needed to confirm this designation. The conclusion is appropriate but should more clearly highlight the novel contributions of this study.”

Response: Thank you for this thoughtful suggestion. We have revised the Discussion section to more explicitly integrate our findings with previous literature on imaging and prognostic differentiation in viral pneumonia. We now highlight how specific radiologic features (e.g., ground-glass opacities, pleural effusion) and clinical scores (SOFA, APACHE IV) inform triage decisions, such as ICU admission and need for mechanical ventilation. We also discuss how differences in ARDS severity and pneumonia type across COVID-19 and H1N1 may influence therapeutic priorities, including early respiratory support and risk stratification for mortality. We have revised the Study Design and Methods section to specify that patient enrollment was consecutive, initiated at the time of hospital admission for respiratory symptoms, and that all imaging and data extraction were conducted prospectively using predefined protocols. These clarifications confirm the prospective nature of the study and ensure transparency regarding participant selection and data handling procedures.

Text added: “Our findings align with previous studies showing that COVID-19 is often characterized by extensive ground-glass opacities and peripheral distribution, while H1N1 tends to present with pleural effusion and consolidation. These radiologic patterns, combined with clinical scoring systems, can support early triage decisions—such as prioritizing ICU-level care for H1N1 patients with high SOFA scores or initiating aggressive pulmonary monitoring in COVID-19 cases with high APACHE IV scores. Identifying these distinct profiles is clinically meaningful for tailoring treatment plans, allocating resources, and optimizing outcomes, especially during periods of overlapping viral circulation.” (Page 15, lines 517-524)

“This was a prospective observational cohort study with consecutive enrollment of eligible patients admitted for severe or critical respiratory symptoms, carried out at Hospital General Regional in León, Mexico, encompassing patients diagnosed with pneumonia caused by either SARS-CoV-2 or influenza A H1N1 between June 2020 and May 2021, a period corresponding to the first and second waves of the COVID-19 pan-demic in Mexico, during which ancestral variants of SARS-CoV-2 predominated prior to the emergence of Omicron-related strains. All CT scans were performed within 24 hours of admission, and clinical and laboratory data were collected prospectively using standardized data abstraction protocols developed prior to patient enrollment.” (Page 3, lines 113-121).

Comment: “Finally, minor inconsistencies in author affiliations, citation formatting, and typographic elements (such as duplicated words or abrupt line breaks) should be corrected before publication.”

Response: Thank you for this helpful observation. We have conducted a thorough review of the manuscript to ensure consistency and accuracy in all formatting elements. Author affiliations have been standardized in line with the journal’s formatting guidelines. Citation styles have been checked and corrected for consistency in superscript numbering and punctuation. Additionally, we have corrected typographic issues such as duplicated words, excessive spacing, and abrupt or misplaced line breaks to improve overall readability and presentation.

We thank the reviewers for their constructive feedback, which has greatly improved the clarity and quality of our manuscript. We have addressed each comment thoroughly and incorporated the suggested changes. We hope the revised version meets the expectations of the reviewers and editorial team, and we remain available for further revisions if necessary.

Sincerely,

Jaime D. Mondragon, M.D., Ph.D.

University Medical Center Groningen

San Diego State University

Universidad Nacional Autónoma de México

Round 2

Reviewer 4 Report

Comments and Suggestions for Authors

Dear Authors,

the manuscript has been revised according to the comments and in this form is acceptable for further consideration for publication.

Thank You for Your extremely detailed answers and revision.

Sincerely